# Limited role for meteorological factors on the variability in COVID-19 incidence: A retrospective study of 102 Chinese cities

Ka Chun Chong[1,2,3], Jinjun Ran[4], Steven Yuk Fai Lau[1], William Bernard Goggins[1], Shi Zhao[1,2], Pin Wang[5], Linwei Tian[4], Maggie Haitian Wang[1,2], Kirran N. Mohammad[1], Lai Wei[1], Xi Xiong[6], Hengyan Liu[7], Paul Kay Sheung Chan[8], Huwen Wang[9], Yawen Wang[1]*, Jingxuan Wang[1]*

1 Jockey Club School of Public Health and Primary Care, The Chinese University of Hong Kong, Hong Kong Special Administrative Region, China, 2 Clinical Trials and Biostatistics Laboratory, Shenzhen Research Institute, The Chinese University of Hong Kong, China, 3 Centre for Health Systems and Policy Research, The Chinese University of Hong Kong, Hong Kong Special Administrative Region, China, 4 School of Public Health, Li Ka Shing Faculty of Medicine, University of Hong Kong, Hong Kong Special Administrative Region, China, 5 Yale School of Public Health, Yale University, New Haven, United States of America, 6 Department of Pharmacology and Pharmacy, Li Ka Shing Faculty of Medicine, University of Hong Kong, Hong Kong Special Administrative Region, China, 7 School of Nursing, Li Ka Shing Faculty of Medicine, University of Hong Kong, Hong Kong Special Administrative Region, China, 8 Department of Microbiology, Faculty of Medicine, Chinese University of Hong Kong, Hong Kong Special Administrative Region, China, 9 School of Public Health, Shanghai Jiao Tong University, Shanghai, China

* 1155149224@link.cuhk.edu.hk (YW); jxwang@link.cuhk.edu.hk (JW)

**Data Availability Statement:** All relevant data are within the manuscript and its Supporting Information files.

## Abstract

While many studies have focused on identifying the association between meteorological factors and the activity of COVID-19, we argue that the contribution of meteorological factors to a reduction of the risk of COVID-19 was minimal when the effects of control measures were taken into account. In this study, we assessed how much variability in COVID-19 activity is attributable to city-level socio-demographic characteristics, meteorological factors, and the control measures imposed. We obtained the daily incidence of COVID-19, city-level characteristics, and meteorological data from a total of 102 cities situated in 27 provinces/municipalities outside Hubei province in China from 1 January 2020 to 8 March 2020, which largely covers almost the first wave of the epidemic. Generalized linear mixed effect models were employed to examine the variance in the incidence of COVID-19 explained by different combinations of variables. According to the results, including the control measure effects in a model substantially raised the explained variance to 45%, which increased by >40% compared to the null model that did not include any covariates. On top of that, including temperature and relative humidity in the model could only result in < 1% increase in the explained variance even though the meteorological factors showed a statistically significant association with the incidence rate of COVID-19. In conclusion, we showed that very limited variability of the COVID-19 incidence was attributable to meteorological factors. Instead, the control measures could explain a larger proportion of variance.

**Funding:** KCC received the funding from National Natural Science Foundation of China (71974165, 31871340) (http://www.nsfc.gov.cn/) and partially from Health and Medical Research Fund (INF-CUHK-1, 19181132) (https://rfs1.fhb.gov.hk/english/funds/funds_hmrf/funds_hmrf_abt/funds_hmrf_abt.html). The funders had no role in study design, data collection and analysis, decision to publish, or preparation of the manuscript.

**Competing interests:** The authors have declared that no competing interests exist.

## Author summary

COVID-19 has a great impact worldwide, especially in some rural settings where health-care resources are not sufficient. While control measures in these area may be limited, scholars have been discussing the potential effects of meteorological factors on mitigating COVID-19 transmission. Unfortunately, the majority of literatures only looked at the association between COVID-19 and environmental factors in which their findings could mislead readers that certain environmental conditions could be 'protective'. In this study, we argue that the impact of the meteorological factors was very limited by using the incidence data from 102 Chinese cities in the first epidemic period when control measures have been taken into account. As what we expected, once the control measures have been incorporated in the modelling analysis, the meteorological factors could only explain < 1% increase in variability of COVID-19 while control measure explained the variance for more than 40% in total. Because of it, we suggest stringent control measures are necessary to control COVID-19 regardless the meteorological conditions of an area. Given that no vaccine is available to date, our investigation provides an additional evidence, as advocated by World Meteorological Organization rather than relying on changes in the natural environment for mitigation, active non-pharmaceutical interventions are necessary to curb the COVID-19 pandemic.

## Introduction

Pneumonia cases associated with a novel coronavirus were first recognised at the end of December 2019 in Wuhan City, Hubei Province of China. The 2019 coronavirus disease (COVID-19) soon spread to all 34 provinces of China by the end of January. In response to this epidemic, a lockdown was imposed in Wuhan city starting from 23 January 2020, followed by travel restrictions in Hubei. By 29 January 2020, a total of 7,711 confirmed cases and 170 deaths were diagnosed in China, and 31 provinces/municipalities had launched the highest level (level I) of response for major public health emergencies [1] which aimed at preventing and controlling the emergency, to guide and standardize emergency-handling strategies and to minimize the harm caused by such emergency in a prompt and effective manner [2]. The control measure strategies covered nine main medical, social and political aspects: direct leading from the State Council, definition of risk areas, screening of the floating population, traffic control, social distancing, resource mobilization, information release, public education and maintaining social stability. Despite the restrictive political and societal interventions, the disease soon becomes a global public health threat. As of 11 March 2020, the World Health Organization (WHO) reported 118,319 confirmed cases and 4,292 deaths in over 100 countries/regions [3] and announced COVID-19 as a global pandemic on the same day [4].

Scholars have been discussing the potential effects of meteorological factors on COVID-19 transmission. Previous influenza studies found that in cold and dry weather, respiratory droplets remain airborne longer, the virus is more stable and hosts tend to have weakened immunity, which therefore facilitate virus transmission [5]. Existing laboratory data also suggested that severe acute respiratory syndrome coronavirus 2 (SARS-CoV-2) was more stable at a low temperature [6]. Yet, among the population-based studies, the meteorological effects were inconsistent [7–12] and none have assessed the extent to which the effect contributed to the variability of COVID-19 incidence. It was argued that the inconsistent findings may be due to the decreasing impacts of meteorological conditions after the imposition of political and societal measures for epidemic control [12].

While many studies have focused on identifying the association between meteorological factors and the activity of COVID-19, we hypothesize that the impact of the meteorological factors on risk of COVID-19 was minimal when the effects of control measures were taken into account. In this study, we aim to use data from the first wave of epidemic in China (outside Hubei) to assess how much variability in COVID-19 activity is attributable to city-level socio-demographic characteristics, meteorological factors, and control measures of level I responses. We argue stringent control measures are necessary to control COVID-19 regardless the meteorological conditions of an area.

## Method

### Ethics statement

The is a statistical modelling study using publicly available data and all the data were in aggregated level without personal information so no ethical issues were encountered.

### Settings and primary data screening

We obtained data from a total of 102 cities situated in 27 provinces/municipalities outside Hubei province in China from 1 January 2020 to 8 March 2020, which covers almost the first wave of the epidemic. These cities were selected given that at least 20 cases of COVID-19 were confirmed during the study period. While the epidemics in the Chinese cities outside Hubei province consistently characterised by a mixture of imported from Hubei and local cases, Hubei was regarded as an epidemic centre that exhibited completely different spatial dissemination and temporal dynamics of COVID-19 [13], and thus, we excluded all cities in Hubei province from the analysis. Fig 1 shows the spatial distribution of the 102 Chinese cities selected in this study.

### Level I responses in different Chinese provinces and municipalities

By 25 January 2020, all 27 provinces in our study had initiated the level I response to major public health emergencies. The schedule of level I responses and corresponding control measures in each province/municipality were summarized in S1 Table. As a follow-up to the provincial response, many cities launched more specific and multi-dimensional measures. For example, closure of museums, tourist area and religious institutes in Hangzhou, Zhejiang Province, forced masks-wearing and body temperature-checking in public spaces in Shenzhen, Guangdong Province, as well as cancelling mass gathering activities and screening for individuals with travel history from Hubei in Quanzhou, Fujian Province.

### COVID-19 surveillance data

Daily number of confirmed cases of COVID-19 in different Chinese cities from 1 January 2020 to 8 March 2020 were obtained from the webpage of the National Health Commissions of the People's Republic of China [14]. We employed daily incidence, which is defined as the number of cases with illness onset on that day divided by the population size (per million population) in a city, to describe the activity of COVID-19.

To adjust for the delay between date of illness onset and date of confirmation of COVID-19 diagnosis, we rebuilt the epidemic curves using the following recurrence equation:

$$n^{onset}_{ij} = n^{onset}_{i}(j) = \sum_{u} \phi(u) n^{confirm}_{i}(j + u) \tag{1}$$

where $n^{onset}_{i}(j)$ and $n^{confirm}_{i}(j)$ are the number of cases with onset day $j$ and the number of cases confirmed on day $j$ in city $i$ respectively, and $\phi(u)$ is the discretized probability density

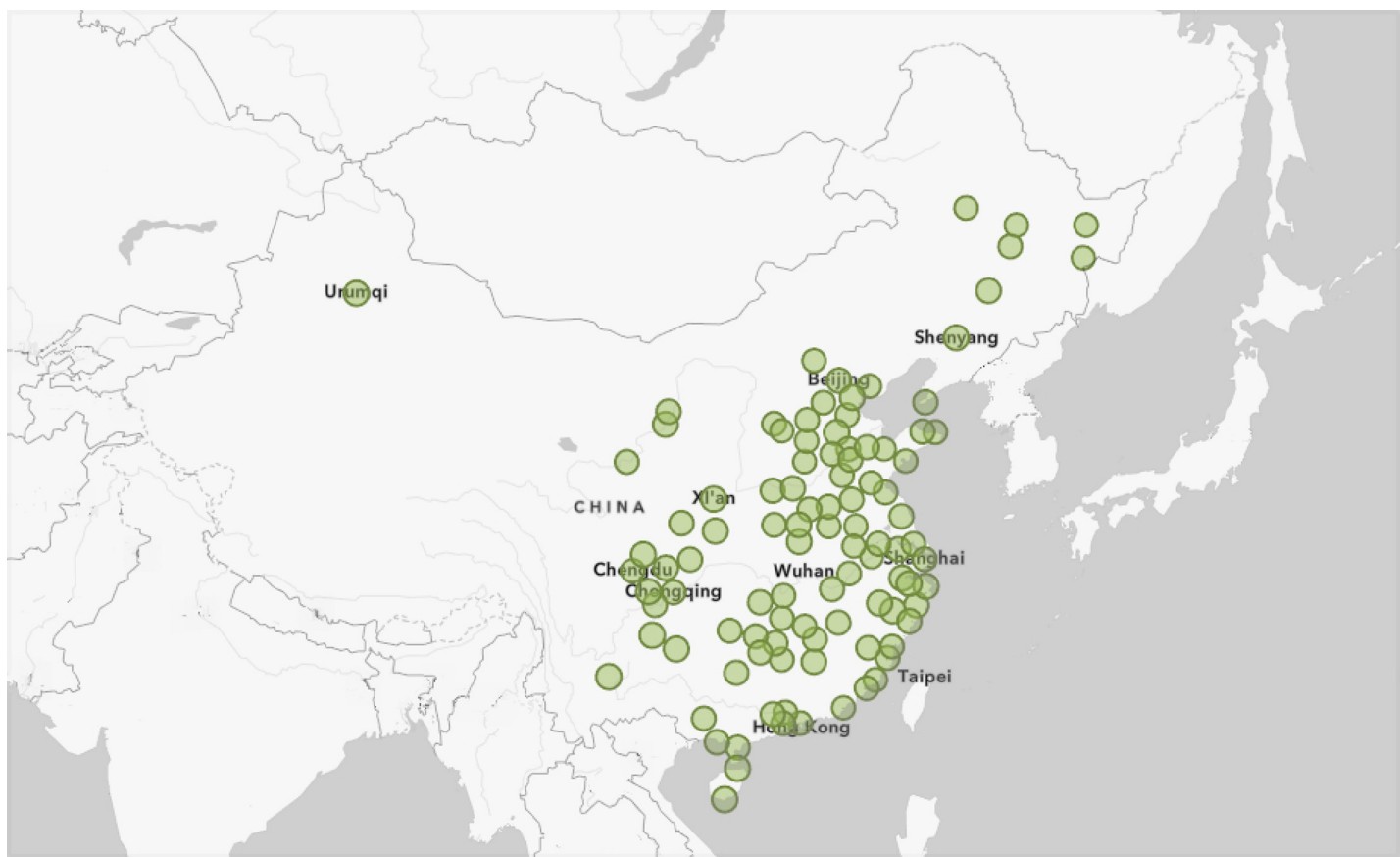

**Fig 1. The 102 Chinese cities selected in the study.**

function of delay duration $U$ that was assumed to follow a gamma distribution with mean of 8.8 days and standard deviation of 4.6 days between 1 and 27 January 2020 and mean of 5.3 days and standard deviation of 3.0 days from 28 January 2020 onwards [13].

## Meteorological data and other covariates

Daily meteorological data including average ambient temperature and relative humidity in each of the cities were collected from the National Climate Data Center [15] and were averaged over all weather stations in a city. As absolute humidity has been demonstrated to have a stronger association with respiratory diseases [16–19], we employed vapour pressure determined by Clausius–Clapeyron equation as a proxy for absolute humidity in our analysis [20–22].

In order to account for the variability between cities in our analysis, we also collected city-specific characteristics including population size, population density, sociodemographic status (i.e. gross domestic product (GDP) per capita, proportion of individuals having tertiary education or above, and proportion of elderly population (i.e. aged >64 years) [23], and geographic distance to Wuhan, which served as a proxy for potential accessibility to the epidemic centre.

## Statistical analysis

To account for between-city variation, we employed generalized linear mixed effect models (GLMMs) to examine the variability in the incidence of COVID-19 explained by different combinations of variables. The GLMMs were fitted using the data from the date of the

epidemic start (i.e. date of having the first case with illness onset) to the date of epidemic end (i.e. date of the last case) in each of the included cities. Suppose $y_{ij}$, the daily incidence rate on day $j$ in city $i$ (i.e. $n^{onset}_{ij}$/population size of city $i$), follows a Poisson distribution with mean $\lambda_{ij}$, the full model form is as follow:

$$\ln(\lambda_{ij}) = \beta_0 + \sum_p \beta_p x_{p_i} + \sum_q \beta_q x_{q_{ij}} + \beta_m x_{m_{ij}} + \beta_t time_i + \alpha_i \qquad (2)$$

where $\beta_0$ is the grand intercept, $x_{p_i}$ is the $p$-th city-specific characteristic variable of city $i$ with regression coefficient $\beta_p$, $x_{q_{ij}}$ is the $q$-th time-varying meteorological variable of city $i$ on day $j$ with regression coefficient $\beta_q$, $x_{m_{ij}}$ is the variable with regression coefficient $\beta_m$ which captures the incremental effect of control measures of level I responses implemented on day $k$ as defined below:

$$x_{m_{ij}} = \begin{cases} 0 & \text{where } j \leq \text{date of control measures implementation} \\ j - k & \text{where } j > \text{date of control measures implementation} \end{cases} \qquad (3)$$

To account for the time trend, we included a variable $time_i$, which is the number of days since the date of the first case with illness in city $i$ with regression coefficient $\beta_t$ in the model. In the GLMM, the city-specific random effect is modelled as $\alpha_i$ which followed a normal distribution with mean 0 and variance $\sigma_\alpha^2$. The use of the random effect is to capture the city-specific heterogeneity that cannot be accounted for by our data. To account for over-dispersion of the outcome variable, $y_{ij}$ was assumed to follow a negative binomial distribution when the standard Pearson chi-squared statistic divided by its residual degree of freedom ($\chi^2/df$) was greater than two.

We compared five regression models: (i) model with time trend only (M1, base model), (ii) model with city-specific characteristics and time trend (M2), (iii) model with city-specific characteristics, meteorological factors, and time trend (M3), (iv) model with city-specific characteristics, control measures variable, and time trend (M4), and (v) model with city-specific characteristics, meteorological factors, control measures variable, and time trend (M5, full model) using R-squares ($R^2$) proposed by Nakagawa and colleagues [24], so as to determine which variable combination best explains the variance of the activity of SARS-CoV-2. We used $R^2_{fixed}$ to depict the proportion of variance explained by the fixed effects and $R^2_{random}$ to depict the proportion of variance explained by the random effects of cities' heterogeneity. $\Delta R^2_{fixed}$ was used to determine the proportion of variance explained by the additional fixed effect terms in each of the M2 to M5 compared with M1. To avoid the problem of collinearity, the impact of vapour pressure was studied in another set of models by replacing temperature and relative humidity with vapour pressure. Relative risks (RR) with corresponding 95% confidence intervals (CIs) and $p$-values ($p$) were employed to quantify the effects of the variables on risk of COVID-19.

A stratified analysis by climate zone was conducted to examine the difference in proportion of variance explained by factors between temperate and subtropical/tropical cities. Of the 102 Chinese cities included, 45 located in the temperate zone and 57 located in the subtropical or tropical zones. Apart from that, we categorized the control measures into 5 types: social distancing, screening and contact tracing, quarantine of risky populations, hospital-related measures, and other public health measures in order to examine the robustness of the composite variable of the level I responses in the GLMM and to assess the statistical significance for each types of the control measures. A similar model form was used (S1 Text).

In the sensitivity analysis, we tested whether adding an interaction term between meteorological factors and control measures in the model would enhance the explained variance. Since

day of week was shown to be associated with the consultation pattern of some non-acute diseases [25, 26], we examined the variability of our results when the day-of-week term was included into the model. We also compared the main results from models using different lags for meteorological factors (i.e. 3 and 7 lag days) to assess the robustness of our findings. All analyses were carried out using software SAS version 9.4.

## Results

Table 1 shows the descriptive statistics of the city-specific characteristics of the selected 102 cities. The population size ranged from around 600 thousand (Sanya, Hainan province) to 34 million (Chongqing municipality), whereas the population density ranged from 66/km$^2$ (Wuzhong, Ningxia province) to 6,523/km$^2$ (Shenzhen, Guangdong province). The GDP per capita and distance to Wuhan ranged from 22 thousand to 190 thousand Chinese yuan and 210 km to 3,270 km respectively. Beijing municipality had the largest proportion of residents with tertiary education (42.3%), while Chongqing had the largest proportion of population aged >64 years (12.9%).

Across all the included cities, the daily ambient temperature and relative humidity ranged from -23.6˚C to 29.5˚C and 9.4% to 100% respectively (Table 1). The overall median of city-specific mean temperature was 6.9˚C (range: -15.0˚C to 22.6˚C) and the median of city-specific mean temperature increased from 4.8˚C (range: -18.5˚C to 22.2˚C) in January 2020 to 10.0˚C (range: -7.4˚C to 24.3˚C) in March 2020 (Fig 2A). The overall median of city-specific mean relative humidity was 74.4% (Range: 44.9% to 89.7%) and the median of city-specific mean relative humidity decreased slightly from 76.3% (range: 51.6% to 90.9%) in January 2020 to 73.8% (range: 30.3% to 97.9%) in March 2020 (Fig 2B).

Before 22 January 2020, most of the cities had a daily incidence rate below 2 per million population (Fig 2C). Before further upsurge of epidemic outbreaks, the Chinese provincial governments have declared level I responses during 23 to 25 January 2020. After that, Shenzhen in Guangdong province experienced the peak daily incidence of 6.9 per million inhabitants on 28 January 2020 among all cities outside Hubei province. A downward trend was observed in the epidemic curve about a week after the implementation of control measures of level I responses.

Table 2 shows the model comparison results. Compared with M1, which solely included the time trend, including city-specific characteristics in the model (M2) could only explain an additional 3.22% of the variance in the incidence rate. Further inclusion of temperature and relative humidity as time-varying fixed effects in the model (M3) boosted the explained variance to 11.8%. However, having the control measure effects included in the GLMM (M4) substantially raised the explained variance to 45.0%, which increased by >40% compared to the

**Table 1. Descriptive statistics of city-specific characteristics and meteorological factors in 102 cities.**

| | Minimum | 25th percentile | Median | 75th percentile | Maximum |
|---|---|---|---|---|---|
| Population (in 10,000) | 60 | 413 | 605 | 825 | 3,397 |
| Population density (in /km$^2$) | 66 | 254 | 514 | 706 | 6,523 |
| GDP per capita (in 10,000 Chinese Yuan) | 2.2 | 3.9 | 6.3 | 8.8 | 19.0 |
| Proportion of individuals having tertiary education (%) | 8.5 | 10.9 | 12.0 | 13.7 | 42.3 |
| Proportion of elderly population (%) | 7.4 | 9.5 | 10.8 | 11.7 | 12.9 |
| Distances to Wuhan (in 100 km) | 2.1 | 6.7 | 9.3 | 12.5 | 32.7 |
| Ambient temperature over all cities (˚C) | -23.6 | 3.6 | 8.8 | 13.7 | 29.5 |
| Relative humidity over all cities (%) | 9.4 | 58.4 | 72.8 | 84.2 | 100.0 |
| Vapour pressure over all cities (hPa) | 0.6 | 4.9 | 7.9 | 11.3 | 29.9 |

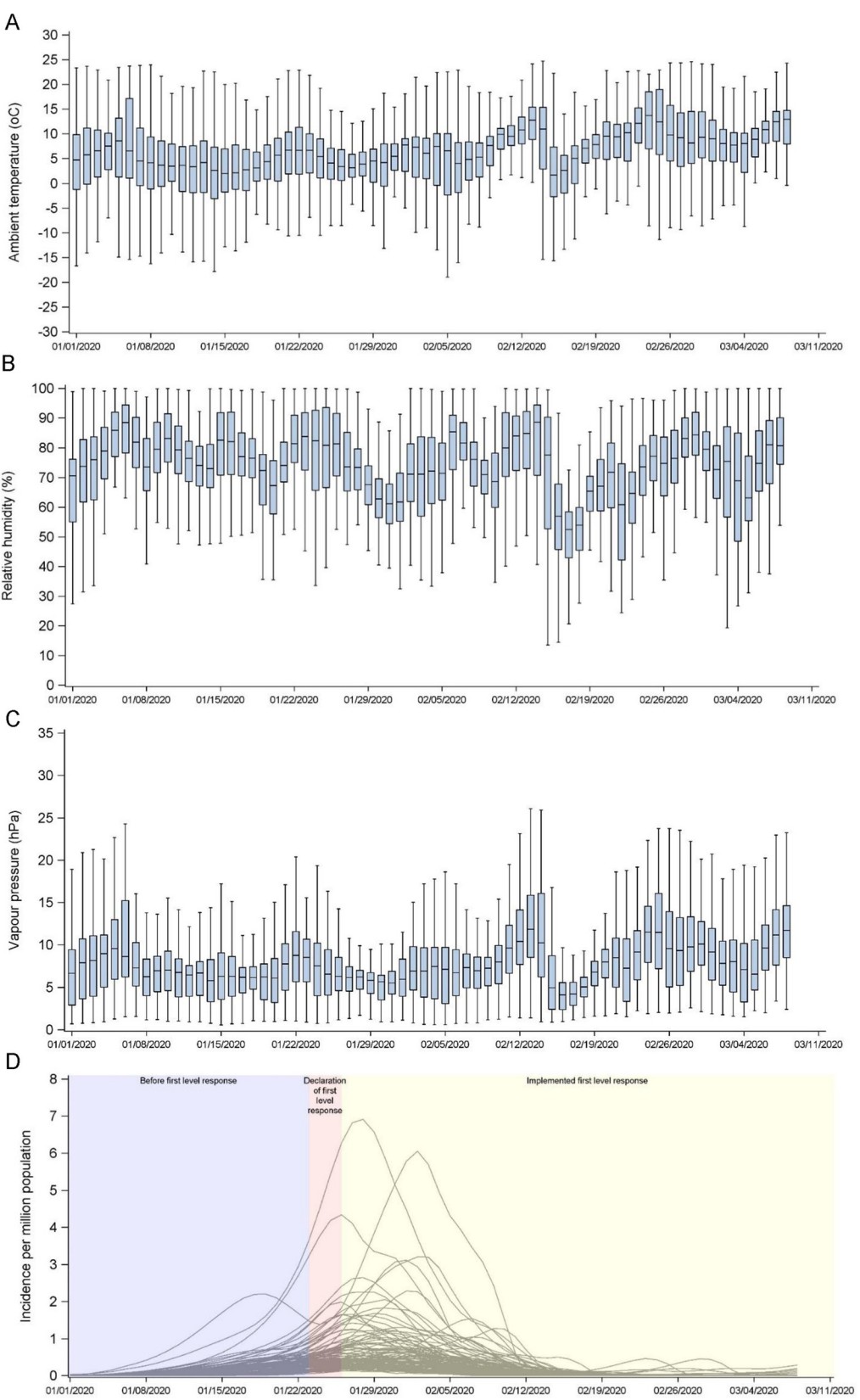

**Fig 2. Temporal distribution of (A) ambient temperature (ºC), (B) relative humidity (%), (C) vapour pressure (hPa), and (D) incidence of COVID-19 infections (per million population).** The level I responses were implemented in different provinces between 23 and 25 January 2020.

null model. On top of this effect, including the meteorological effects in the model (M5) only resulted in < 1% increase in the explained variance even though temperature and relative humidity showed a statistically significant association with the incidence rate of COVID-19 (temperature: RR = 0.984, 95% CI: 0.969–0.999, $p$ = 0.040; relative humidity: RR = 0.993, 95% CI: 0.988–0.997, $p$ = 0.001). In the full model, no city-specific characteristics (i.e. distance to Wuhan, population density, GDP per capita, proportion of tertiary education, and proportion of elderly population) were significantly associated with the COVID-19 incidence.

When temperature and relative humidity in the models were replaced with vapour pressure, the increases in explained variance were similar (Table 3). Nevertheless, a decrease in vapour pressure was statistically significantly associated with an increased risk of the COVID-19 incidence (RR = 0.958, 95% CI: 0.939–0.976, $p$<0.001).

While the analysis was stratified by climate zone, the additional variances explained by the control measures were similar between the temperate and subtropical/tropical cities when compared with the variance explained in the model without the effects of M3 control measure (Table 4). However, the contribution of meteorological factors in the explained variance of the subtropical/tropical cities was around 3-fold more than that in the temperate cities (i.e. $\Delta R^2_{fixed}$ = 14.4% vs $\Delta R^2_{fixed}$ = 5.04% in M3). The temperature and relative humidity even became statistically insignificant in the full model when fitting the data of temperate cities

**Table 2. Comparison of changes in R-square among different models and relative risks (95% confidence intervals) of the variables.**

| Variables | M1 (null model) | M2 | M3 | M4 | M5 (full model) |
|---|---|---|---|---|---|
| City-specific characteristics | | | | | |
| Population density (in /100 km$^2$) | | 1.011 (0.990, 1.033) | 1.022 (0.996, 1.048) | 1.016 (0.994, 1.039) | 1.019 (0.995, 1.043) |
| GDP per capita (in 10,000 Chinese Yuan) | | 1.032 (0.981, 1.084) | 1.051 (0.991, 1.115) | 1.015 (0.993, 1.070) | 1.021 (0.966, 1.078) |
| Proportion of tertiary education (in %) | | 1.003 (0.967, 1.040) | 0.984 (0.942, 1.028) | 1.008 (0.970, 1.048) | 1.000 (0.961, 1.042) |
| Proportion of elderly population (in %) | | 0.927 (0.842, 1.021) | 0.865 (0.771, 0.970)* | 0.921 (0.832, 1.019) | 0.907 (0.815, 1.009) |
| Distances to Wuhan (in 100 km) | | 1.004 (0.979, 1.029) | 0.961 (0.932, 0.992)* | 1.001 (0.975, 1.028) | 0.986 (0.958, 1.016) |
| Meteorological factors | | | | | |
| Temperature (in ºC) | | | 0.943 (0.929, 0.956)** | | 0.984 (0.969, 0.999)* |
| Relative humidity (in %) | | | 0.992 (0.988, 0.996)** | | 0.993 (0.988, 0.997)* |
| Control measure effect | | | | 0.755 (0.739, 0.771)** | 0.755 (0.739, 0.771)** |
| Time trend | 1.009 (1.006, 1.013)** | 1.009 (1.005, 1.012)** | 1.011 (1.007, 1.015)** | 1.227 (1.206, 1.248)** | 1.226 (1.205, 1.248)** |
| $\chi^2/df$ | 0.33 | 0.33 | 0.30 | 0.11 | 0.11 |
| $R^2_{fixed}$ | 0.98% | 4.20% | 11.8% | 45.0% | 45.7% |
| $R^2_{random}$ | 22.7% | 20.9% | 25.6% | 13.3% | 14.0% |
| $\Delta R^2_{fixed}$ | - | 3.22% | 10.8% | 44.0% | 44.7% |

Note: M1, Model with time only; M2, Model with city-specific characteristics and time; M3, Model with city-specific characteristics, meteorological factors, and time; M4, Model with city-specific characteristics, control measure variable, and time; M5, Model with city-specific characteristics, meteorological factors, control measure variable, and time (full model). RR, Relative risk in incidence rate of COVID-19 for each unit change of variable; $\chi^2/df$, chi-square statistics divided by the degree of freedom; $R^2_{fixed}$, Proportion of variance in the incidence rate (per million population) explained by the fixed effect terms; $R^2_{random}$, Proportion of variance explained by the random effect term of cities' heterogeneity. $\Delta R^2_{fixed}$, $R^2_{fixed}$ of each model minus $R^2_{fixed}$ of M1.

*$p$<0.05

**$p$<0.001.

**Table 3. Comparison of changes in R-square among different models and relative risks (95% confidence intervals) of the variables when vapour pressure was used to replace temperature and relative humidity.**

| Variables | M1 (null model) | M3 | M5 (full model) |
|---|---|---|---|
| City-specific characteristics | | | |
| Population density (in /100 km$^2$) | | 1.021 (0.996, 1.047) | 1.020 (0.996, 1.045) |
| GDP per capita (in 10,000 Chinese Yuan) | | 1.043 (0.984, 1.106) | 1.019 (0.965, 1.077) |
| Proportion of tertiary education (in %) | | 0.986 (0.944, 1.029) | 1.001 (0.961, 1.042) |
| Proportion of elderly population (in %) | | 0.865 (0.773, 0.969)* | 0.894 (0.804, 0.995)* |
| Distances to Wuhan (in 100 km) | | 0.988 (0.960, 1.017) | 0.994 (0.967, 1.022) |
| Meteorological factors | | | |
| Absolute humidity (in hPa) | | 0.905 (0.888, 0.923)** | 0.958 (0.939, 0.976)** |
| Control measure effect | | | 0.757 (0.741, 0.773)** |
| Time trend | 1.009 (1.006, 1.013)** | 1.010 (1.007, 1.014)** | 1.225 (1.204, 1.247)** |
| $\chi^2/df$ | 0.33 | 0.30 | 0.11 |
| $R^2_{fixed}$ | 0.98% | 10.9% | 45.6% |
| $R^2_{random}$ | 22.7% | 25.5% | 14.2% |
| $\Delta R^2_{fixed}$ | - | 9.93% | 44.6% |

Note: M1, Model with time only; M3, Model with city-specific characteristics, vapour pressure, and time; M5, Model with city-specific characteristics, vapour pressure, control measure variable, and time (full model). RR, Relative risk in incidence rate of COVID-19 for each unit change of variable; $\chi^2/df$, chi-square statistics divided by the degree of freedom; $R^2_{fixed}$, Proportion of variance in the incidence rate (per million population) explained by the fixed effect terms; $R^2_{random}$, Proportion of variance explained by the random effect term of cities' heterogeneity. $\Delta R^2_{fixed}$, $R^2_{fixed}$ of each model minus $R^2_{fixed}$ of M1.

*$p<0.05$

**$p<0.001$.

(temperature: RR = 0.981, 95% CI: 0.952–1.010, $p$ = 0.198; relative humidity: RR = 0.997, 95% CI: 0.990–1.004, $p$ = 0.458).

When the control measures were categorized by types, the overall variance explained in the models was reduced by around 8% (Table 5). However, as with the major finding, including the meteorological effects in the model (M5) only resulted in 2% increase in the explained variance with the statistical significances of the temperature and relative humidity remain unchanged. Among all types of control measure, imposing social distancing (RR = 0.912, 95% CI: 0.892–0.932, $p<0.001$), screening and contact tracing (RR = 0.945, 95% CI: 0.926–0.965, $p<0.001$), hospital-related measures (RR = 0.954, 95% CI: 0.941–0.967, $p<0.001$), and other public health measures (RR = 0.942, 95% CI: 0.927–0.958, $p<0.001$) were significantly associated with a lower risk of COVID-19. Quarantine of risky populations was not found to be a significant predictor.

As shown in the sensitivity analysis, our results were robust to variance explained by the delayed effects of meteorological factors (S2 Table). When the lags of temperature and relative humidity were increased in the models, a slight decrease in the explained variance was observed (lag = 3 days: $R^2_{fixed}$ = 10.4% in M3; lag = 7 days: $R^2_{fixed}$ = 9.15% in M3) and both of the temperature and relative humidity tended to be less significant. Compared with $R^2_{fixed}$ of M4, $R^2_{fixed}$ of the full model that accounted for lag effects did not change remarkably and was kept at around 45%. Adding the interaction terms (S3 Table) or the day-of-week term (S4 Table) in the model did not enhance the proportion of variance explained.

**Table 4. Comparison of changes in R-square among different models and relative risks (95% confidence intervals) of the variables by climate zone.**

| Climate zone | Variables | M1 (null model) | M3 | M5 (full model) |
|---|---|---|---|---|
| Temperate | City-specific characteristics | | | |
| | Population density (in /100 km$^2$) | | 0.971 (0.880, 1.072) | 0.959 (0.854, 1.077) |
| | GDP per capita (in 10,000 Chinese Yuan) | | 0.975 (0.904, 1.053) | 0.946 (0.865, 1.034) |
| | Proportion of tertiary education (in %) | | 1.031 (0.988, 1.076) | 1.046 (0.994, 1.100) |
| | Proportion of elderly population (in %) | | 0.914 (0.787, 1.060) | 0.923 (0.777, 1.097) |
| | Distances to Wuhan (in 100 km) | | 0.974 (0.934, 1.016) | 0.987 (0.940, 1.037) |
| | Meteorological factors | | | |
| | Temperature (in °C) | | 0.959 (0.934, 0.984)* | 0.981 (0.952, 1.010) |
| | Relative humidity (in %) | | 0.996 (0.990, 1.003) | 0.997 (0.990, 1.004) |
| | Control measure effect | | | 0.771 (0.745, 0.798)** |
| | Time trend | 1.015 (1.009, 1.021)** | 1.018 (1.011, 1.026)** | 1.219 (1.184, 1.254)** |
| | $\chi^2/df$ | 0.24 | 0.23 | 0.09 |
| | $R^2_{fixed}$ | 2.40% | 7.43% | 43.1% |
| | $R^2_{random}$ | 15.1% | 14.2% | 12.1% |
| | $\Delta R^2_{fixed}$ | - | 5.04% | 40.7% |
| Subtropical/tropical | City-specific characteristics | | | |
| | Population density (in /100 km$^2$) | | 1.018 (0.988, 1.048) | 1.014 (0.989, 1.039) |
| | GDP per capita (in 10,000 Chinese Yuan) | | 1.083 (1.002, 1.170)* | 1.071 (1.004, 1.142)* |
| | Proportion of tertiary education (in %) | | 0.957 (0.876, 1.045) | 0.967 (0.899, 1.041) |
| | Proportion of elderly population (in %) | | 0.866 (0.741, 1.012) | 0.924 (0.810, 1.053) |
| | Distances to Wuhan (in 100 km) | | 1.024 (0.957, 1.095) | 0.966 (0.913, 1.022) |
| | Meteorological factors | | | |
| | Temperature (in °C) | | 0.901 (0.881, 0.921)** | 0.979 (0.956, 1.002) |
| | Relative humidity (in %) | | 0.991 (0.985, 0.996)* | 0.989 (0.984, 0.994)** |
| | Control measure effect | | | 0.744 (0.724, 0.765)** |
| | Time trend | 1.006 (1.002, 1.011)* | 1.007 (1.003, 1.012)* | 1.233 (1.206, 1.260)** |
| | $\chi^2/df$ | 0.36 | 0.34 | 0.13 |
| | $R^2_{fixed}$ | 0.48% | 14.8% | 51.3% |
| | $R^2_{random}$ | 27.8% | 27.0% | 11.3% |
| | $\Delta R^2_{fixed}$ | - | 14.4% | 50.8% |

Note: M1, Model with time only; M3, Model with city-specific characteristics, meteorological factors, and time; M5, Model with city-specific characteristics, meteorological factors, control measure variable, and time (full model). RR, Relative risk in incidence rate of COVID-19 for each unit change of variable; $\chi^2/df$, chi-square statistics divided by the degree of freedom; $R^2_{fixed}$, Proportion of variance in the incidence rate (per million population) explained by the fixed effect terms; $R^2_{random}$, Proportion of variance explained by the random effect term of cities' heterogeneity. $\Delta R^2_{fixed}$, $R^2_{fixed}$ of each model minus $R^2_{fixed}$ of M1.

*$p < 0.05$

**$p < 0.001$.

## Discussion

Although laboratory findings showed that the stability of SARS-CoV-2 was sensitive to temperature and relative humidity [6,27,28] in controlled environments, the meteorological effect varies greatly at population level when host factors were taken into account. In this study, we employed data from 102 Chinese cities which experienced the first wave of epidemic to assess how much variability in COVID-19 activity was attributable to city-level socio-demographic characteristics, meteorological factors, and control measures of level I responses. According to our results, despite temperature and relative humidity were significantly associated with the risk of COVID-19, we could not identify a substantial variability of the COVID-19 incidence

**Table 5. Comparison of changes in R-square among different models and relative risks (95% confidence intervals) of the variables with control measures stratified by type.**

| Variables | M4 | M5 (full model) |
|---|---|---|
| City-specific characteristics | | |
| Population density (in /100 km$^2$) | 1.017 (0.990, 1.044) | 1.022 (0.994, 1.051) |
| GDP per capita (in 10,000 Chinese Yuan) | 1.031 (0.970, 1.095) | 1.040 (0.974, 1.109) |
| Proportion of tertiary education (in %) | 0.984 (0.941, 1.030) | 0.945 (0.929, 1.023) |
| Proportion of elderly population (in %) | 0.951 (0.845, 1.069) | 0.921 (0.812, 1.045) |
| Distances to Wuhan (in 100 km) | 1.009 (0.979, 1.040) | 0.985 (0.952, 1.019) |
| Meteorological factors | | |
| Temperature (in $^o$C) | | 0.970 (0.955, 0.986)* |
| Relative humidity (in %) | | 0.993 (0.989, 0.997)* |
| Control measure effect | | |
| Social distancing | 0.915 (0.896, 0.935)** | 0.912 (0.892, 0.932)** |
| Screening and contact tracing | 0.942 (0.923, 0.960)** | 0.945 (0.926, 0.965)** |
| Quarantine of risky populations | 1.007 (0.993, 1.022) | 1.009 (0.994, 1.024) |
| Hospital-related measures | 0.954 (0.942, 0.967)** | 0.954 (0.941, 0.967)** |
| Other public health measures | 0.942 (0.927, 0.957)** | 0.942 (0.927, 0.958)** |
| Time trend | 1.161 (1.146, 1.175)** | 1.161 (1.146, 1.176)** |
| $\chi^2/df$ | 0.16 | 0.15 |
| $R^2_{fixed}$ | 37.0% | 38.7% |
| $R^2_{random}$ | 19.2% | 20.5% |
| $\Delta R^2_{fixed}$ | 36.0% | 37.7% |

Note: M4, Model with city-specific characteristics, different control measures, and time; M5, Model with city-specific characteristics, meteorological factors, different control measures, and time (full model). RR, Relative risk in incidence rate of COVID-19 for each unit change of variable; $\chi^2/df$, chi-square statistics divided by the degree of freedom; $R^2_{fixed}$, Proportion of variance in the incidence rate (per million population) explained by the fixed effect terms; $R^2_{random}$, Proportion of variance explained by the random effect term of cities' heterogeneity. $\Delta R^2_{fixed}$, $R^2_{fixed}$ of each model minus $R^2_{fixed}$ of M1 in Table 1.

*$p<0.05$

**$p<0.001$.

was attributable to meteorological factors once the effect of control measures of level I response was taken into account. Instead, the implementation of control measures was associated with a larger proportion of variance explained with regard to the activity of COVID-19 and the result was robust to variations in climate zones of the cities and lag effects of meteorological factors.

Our findings support that control measures have significant effects on COVID-19 incidences while climatic conditions are less important in the limits of this study. This corroborates with an investigation by te Beest and colleagues [29] which focused on seasonal influenza, another respiratory disease with likely identical transmission route (via contact, droplets, and fomites). They [29] showed that the effect of absolute humidity could only explain very limited proportion of variance in disease transmission intensity, instead, depletion of susceptible during an epidemic that might be done by vaccination contributed to one-third of total variance. Our study suggests a similar perspective that the effect of host factors likely contributes much variability to COVID-19 transmission at population level even though laboratory findings suggested the viral spreading ability of coronavirus reduced in hot condition [30]. This, on the other hand, suggests that if a vaccine is not available, non-

pharmaceutical interventions to reduce the frequency of host contacts (e.g. social distancing) are required to induce a decrease in COVID-19 incidence.

Although we could not show a large proportion of variance explained by the meteorological factors, temperature and relative humidity were negatively associated with the risk of COVID-19. Another investigation in China also indicated that temperature was a driver of the COVID-19 outbreak and the incidence decreased with the rise of temperature [31]. Consistent with a study in the United States [7], higher temperature was significantly associated with a linearly decreasing risk of COVID-19. Our study echoed with a recent systematic review that hot and wet climates were related to a decrease in spread of COVID-19 [32]. Nevertheless, the association identified in our study contradicts the results of an earlier study showing that high temperature favoured the transmission of COVID-19 in Brazil [33]. Yet, we noted that the Brazil study did not account for the increase in intensity of control measures along time. Such inconsistency of association between temperature and disease transmission was typically observed in respiratory diseases across different zones and hemispheres [13]. We also showed relative humidity and absolute humidity were correlated with the activity of COVID-19 and the result was consistent with other studies [7,33]. We speculate COVID-19 shares similar viral characteristics with influenza in which a lower humidity level could enhance the survival and transmission of the virus [18].

In our analysis, we employed random effects to capture the city-specific heterogeneity that cannot be accounted for by our data. Random effects term, together with the fixed effects terms, helped to increase the variance explained by the models to around 60% of the total variability. The remaining unexplained variance could be attributed to many other factors. For example, we did not capture the between-province heterogeneity which might be inherited from the variation in the compliance of the control measures in level I response (S1 Table). Nevertheless, we conducted an additional analysis by including different types of control measures in the GLMMs and the results were consistent with our major finding though a decrease of model fitness was observed. We also found majority of control measures (i.e. social distancing, screening and contact tracing, hospital-related measures, and other public health measures) was significantly associated with a lower risk of COVID-19 activity. Moreover, different levels of reporting rates may contribute to the unexplained variance, especially when the public awareness of the newly emerged COVID-19 has increased compared to the start of the epidemic which might only be partly captured by the time effect.

There are several major limitations in our study. First, we did not account for the changes in number of susceptible individuals by taking it as one of the fixed effects in our statistical models. However, given COVID-19 is a newly emerged infectious disease, the effect of variation in number of susceptibles on our results is likely to be comparatively minor [29]. Second, we did not study the impacts of other meteorological variables such as rainfall because majority of studies have only documented the impacts of temperature and humidity. Ultraviolet radiation was also shown to be insignificantly associated with the transmission of COVID-19 [10]. Moreover, since the pollution level in China was likely to be reduced at the moment due to shutdown of business and industrial activities, pollutants were not included in our analysis so as to avoid interpretation of non-causal relationship [34]. However, we cannot completely rule out a potential effect of pollutants on exacerbating the prognosis of COVID-19 especially in elderly with chronic conditions as we observed more cases and deaths in the elderly [35–37]. In addition, our study period covered the first wave of COVID-19 epidemic which only lasted for three months. Our findings may thus not be generalized to other seasons although our study period covered a wide range of meteorological variation in most Chinese cities across a year. Third, we used a single variable to capture the effect of control measures of level I response in the model due to complexity in differentiating the impacts of each control

measure. Further modelling investigation using more information to rank the importance of factors in explaining the reduction of COVID-19 incidence is warranted.

In conclusion, even though meteorological factors were associated with COVID-19, we could not find an apparent impact of them and only the effect of control measures could explain a large portion of variability in COVID-19 activity. Therefore, we argue stringent control measures are necessary to control COVID-19 regardless the meteorological conditions of an area. Given that no vaccine is available to date, our investigation provides an additional evidence, as advocated by World Meteorological Organization [38], rather than relying on changes in the natural environment for mitigation, active non-pharmaceutical interventions are necessary to curb the COVID-19 pandemic.

## Supporting information

**S1 Table. Summary of control measures of Level I response in 27 provinces/municipalities.**
(DOCX)

**S2 Table. Comparison of changes in R-square among different models and relative risks (95% confidence intervals) of the variables when different lags are selected for meteorological factors.**
(DOCX)

**S3 Table. Changes of R-square and relative risks (95% confidence intervals) of the variables when interaction terms were included.**
(DOCX)

**S4 Table. Changes of R-square and relative risks (95% confidence intervals) of the variables when day-of-week was included.**
(DOCX)

**S1 Text. Generalized linear mixed effect models (GLMMs) for different types of control measures.**
(DOCX)

## Acknowledgments

We thank Dr. Tsz Cheung Lee from the Hong Kong Observatory for his assistance in the interpretation of study findings.

## Author Contributions

**Conceptualization:** Ka Chun Chong, Steven Yuk Fai Lau, William Bernard Goggins, Shi Zhao, Pin Wang, Jingxuan Wang.

**Data curation:** Ka Chun Chong, Jinjun Ran, Yawen Wang, Jingxuan Wang.

**Formal analysis:** Ka Chun Chong, Steven Yuk Fai Lau, William Bernard Goggins, Lai Wei, Xi Xiong, Hengyan Liu, Jingxuan Wang.

**Funding acquisition:** Ka Chun Chong, Maggie Haitian Wang.

**Investigation:** Ka Chun Chong, Kirran N. Mohammad, Lai Wei, Jingxuan Wang.

**Methodology:** Ka Chun Chong, Steven Yuk Fai Lau, William Bernard Goggins, Shi Zhao, Pin Wang, Kirran N. Mohammad, Paul Kay Sheung Chan, Huwen Wang, Jingxuan Wang.

**Project administration:** Ka Chun Chong, Maggie Haitian Wang, Paul Kay Sheung Chan.

**Resources:** Ka Chun Chong.

**Software:** Ka Chun Chong, Lai Wei, Jingxuan Wang.

**Supervision:** Ka Chun Chong, Linwei Tian, Paul Kay Sheung Chan.

**Visualization:** Kirran N. Mohammad, Xi Xiong, Hengyan Liu.

**Writing – original draft:** Ka Chun Chong, Kirran N. Mohammad, Huwen Wang, Jingxuan Wang.

**Writing – review & editing:** Ka Chun Chong, Jinjun Ran, Steven Yuk Fai Lau, William Bernard Goggins, Shi Zhao, Pin Wang, Linwei Tian, Maggie Haitian Wang, Xi Xiong, Hengyan Liu, Paul Kay Sheung Chan, Huwen Wang, Yawen Wang.

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
