## [Decision Letter · Decision Letter 0]

27 Oct 2020

Dear Dr. Chong,

Thank you very much for submitting your manuscript "Limited role for meteorological factors on the variability in COVID-19 incidence: A retrospective study of 102 Chinese cities" for consideration at PLOS Neglected Tropical Diseases. As with all papers reviewed by the journal, your manuscript was reviewed by members of the editorial board and by several independent reviewers. In light of the reviews (below this email), we would like to invite the resubmission of a significantly-revised version that takes into account the reviewers' comments. 

We cannot make any decision about publication until we have seen the revised manuscript and your response to the reviewers' comments. Your revised manuscript is also likely to be sent to reviewers for further evaluation.

Sincerely,

Piet Maes, Ph.d.

Deputy Editor

Victor Santos

Deputy Editor

Reviewer's Responses to Questions

**Key Review Criteria Required for Acceptance?**

**Methods**

-Are the objectives of the study clearly articulated with a clear testable hypothesis stated?

-Is the study design appropriate to address the stated objectives?

-Is the population clearly described and appropriate for the hypothesis being tested?

-Is the sample size sufficient to ensure adequate power to address the hypothesis being tested?

-Were correct statistical analysis used to support conclusions?

-Are there concerns about ethical or regulatory requirements being met?

Reviewer #1: I wonder if the procedure for incorporating the incremental effect of control measures of level I responses in the statistical model does not exaggerate this factor too much. Xmij in Equation 3 grows more and more positive after the date of control measure implementation. This way possible temporal variability in the implementation of the measures is not considered. 

On the city-specific random effect: differences in local measures and the way they are applied are very important and must be accounted for somehow in this random effect. How is this done in this research? Please elaborate on this. How sensitive is this choice for the overall result? 

Experiences show that the day of the week affects data collection. At some days in the week people tend to visit the doctor more frequently for testing than on other days. So why is this variable not considered in the statistical model?

Reviewer #2: See the 'Summary and General Conclusions' section

**Results**

-Does the analysis presented match the analysis plan?

-Are the results clearly and completely presented?

-Are the figures (Tables, Images) of sufficient quality for clarity?

Reviewer #1: Explain the ** in the tables;

Are the boxplots showing the max/min values or the 95/5% values?

Reviewer #2: See the 'Summary and General Conclusions' section

**Conclusions**

-Are the conclusions supported by the data presented?

-Are the limitations of analysis clearly described?

-Do the authors discuss how these data can be helpful to advance our understanding of the topic under study?

-Is public health relevance addressed?

Reviewer #1: The conclusion of this manuscript is perhaps not that meteorology has few impact on COVID19, but that the impact of meteorology could not be assessed due to the huge impact of the control measures, which is clearly suggested in the title (fine for me!), but some conclusions should be weakened more in the manuscript.

Reviewer #2: See the 'Summary and General Conclusions' section

**Editorial and Data Presentation Modifications?**

Reviewer #1: NA

Reviewer #2: See the 'Summary and General Conclusions' section

**Summary and General Comments**

Reviewer #1: Overall an interesting and relevant paper, covering a contemporary topic. The manuscript is fairly well written, and the applied basic statistical methodology seems to be sound enough. My two main concerns are that these data sets only cover a very short period so that the effect of meteorological factors cannot be properly identified since the meteorological variation seems not very large (based on what I could see from the graphs in Fig 2), and since the control measures that were taken are so draconic that these eliminate all other possible factors. Meteorological factors affecting COVID19 activity are just drawn by the control measures. Let me be clear, I am not judging the control measures that were taken.

Reviewer #2: The authors present the results of an analysis on the determinants of COVID-19 over more than 100 Chinese cities. The tested variables belong to three major groups: socio-demographic aspects, meteorology and control measures. 

The study is relevant as it contributes to the current global efforts towards understanding the drivers of the spread of the virus. Therefore, I would recommend its publication provided a number of remarks/questions are clarified. Especially the remarks 2 to 6 concern aspects of the study that – I consider- deserve more elaboration or argumentation from the authors.

1. The caption of Figure 1 is “The 102 Chinese cities selected in the study”. The Figure, however, seems to represent municipalities or provinces instead of cities. Is that the case? Are the colored shapes in the map actually showing the 27 provinces (Line 116) where the study was conducted? Moreover, the Figure intends to represent a map but no geographical references (like grid with coordinates, North arrow, scale) are given. It would also be useful to show labels indicating the location of (at least) the main places mentioned in the manuscript (Hubei, Ningxia, Guangdong, etc). In its current state the Figure 1 is not really helpful to the reader who is not familiar with the geography in that part of the world. I would also recommend to write the full word ‘Figure’ instead of ‘Fig’ in the body of the text and the figure caption.

2. Equation 2 indicates that the time step of the analysis is 1 day. However, the variables in the model related to city-specific characteristics (population density, GDP, distance to Wuhan, etc) do not change daily, by nature; in fact, they remain unchanged during the studied period. For each particular city they behave as constants in the model as compared to other time-varying potential drivers like meteorology and control measure effects. What can be the impact of combining time-varying and constant drivers of the spread in the model of Equation 2. Was that a good approach? Could that be the explanation of the small difference between the results of M1 and those of M2, as explained in Lines 227-228? Could you elaborate more on that?

3. Lines 165-166 explain the role of the timei variable. This variable seems to be quite arbitrary and only dependent on the date chosen by the researchers to start the study. Therefore, not really independent. How can this decision be justified? And what about the collinearity between the timei and the xmij variable (Equation 3)? I guess the value of those variables is quite similar as they are built on the same manner: starting from 0 and adding 1 every elapsed day.

4. The values of the variable xmij as indicated in Equation 3 are only dependent on the date of implementation of the control measures. Does it mean an implicit assumption that the impact of the different measures listed in Table S1 is equivalent (i.e. the efficiency of the control measures in the 27 provinces is comparable)? 

5. One of the important conclusions of the study is that “the impact of meteorological factors was very limited” (Discussions section). The considered period of the study was, however, quite short and covered only one season; i.e. only a (small) fraction of the annual range of variation of temperature and humidity was investigated. Could we be confident that such a conclusion will hold in other seasons of the year too? If not, it should be mentioned in the text. If I understood well, the study did not investigate whether the incidence was somehow related to climatic zones; we do not know whether the incidence is higher/lower in dry/humid warm/cold regions. The box plots in Figures 2A, 2B, 2C show, however, that the spatial variability is considerable. Such analysis would strongly support the conclusion stated in the manuscript.

6. The study concludes that the impact of meteorology in the incidence of COVID 19 is limited. In addition to that, it is indicated in lines 227-228 that city-specific characteristics do not explain much of the variance in incidence rate. The control measures were launched more or less at the same time. What is then the main driver of the different incidence patterns observed in the studied cities and shown in Figure 2D? What is really driving the incidence? Has your model been available to detect that? The conclusion (last paragraph of the manuscript) indicates that “only the effect of control measures could explain a large portion of variability in COVID-19 activity”. Can we conclude then that the different patterns (phase and amplitude) observed in Figure 2D are basically explained by different levels of efficiency of the control measures (and the random aspect)?

PLOS authors have the option to publish the peer review history of their article (what does this mean?). If published, this will include your full peer review and any attached files.

Reviewer #1: No

Reviewer #2: No
---

## [Editor Report · Decision Letter 1]

22 Dec 2020

Dear Dr. Chong,

We are pleased to inform you that your manuscript 'Limited role for meteorological factors on the variability in COVID-19 incidence: A retrospective study of 102 Chinese cities' has been provisionally accepted for publication in PLOS Neglected Tropical Diseases.

Best regards,

Piet Maes, Ph.d.

Deputy Editor

Victor Santos

Deputy Editor

We apologize for the confusion - the initial decision sent was a mistake and we intended to accept this paper for publication. Please let us know if you have any questions and we apologize for the error.

---

## [Editor Report · Acceptance letter]

7 Feb 2021

Dear Dr. Chong,

We are delighted to inform you that your manuscript, "Limited role for meteorological factors on the variability in COVID-19 incidence: A retrospective study of 102 Chinese cities," has been formally accepted for publication in PLOS Neglected Tropical Diseases.

Best regards,

Shaden Kamhawi

co-Editor-in-Chief

Paul Brindley

co-Editor-in-Chief
